# Intermittent Use of Anti-Hormonal Agents for the Endocrine Therapy of Sex-Hormone-Dependent Breast and Prostate Cancer: A Protocol for a Systematic Review

**DOI:** 10.3390/ijerph192315486

**Published:** 2022-11-22

**Authors:** Dorothea Kesztyüs, Johanna Kämpfer, Tibor Kesztyüs

**Affiliations:** Medical Data Integration Centre, Department of Medical Informatics, University Medical Centre, Georg-August University Göttingen, 37073 Göttingen, Germany

**Keywords:** breast cancer, prostate cancer, endocrine therapy, intermittent administration

## Abstract

Therapies with the continuous administration of anti-hormonal agents in sex-hormone-dependent malignancies such as prostate and breast carcinomas often lead to the development of resistant tumor cells. A systematic evaluation of the use and effects of the intermittent application of endocrine therapy could provide information on the state of knowledge in this research area. PubMed, Cochrane Library, Embase, and Web of Science will be systematically searched using pretested search strategies. Randomized and non-randomized controlled trials, pragmatic trials, case–control, and comparative cohort studies will be eligible. Primary outcomes will be progression-free survival, disease-free survival, and overall survival. The literature retrieved will be selected based on predefined inclusion and exclusion criteria. Relevant data will be extracted from included references into a pre-designed table. The risk of bias will be assessed, and the report of the results will follow PRISMA recommendations and include any deviations from this protocol. The increasing prevalence of breast and prostate cancer and limitations of current therapeutic approaches require a closer look at alternatives. Additionally, to explore new therapeutic agents, modalities of administration should be rigorously reviewed to determine the best regimens for patients. This proposed systematic review aims to summarize and evaluate the current knowledge regarding intermittent endocrine cancer therapy to provide a basis for further research.

## 1. Introduction

In high-income countries, a transition is emerging from cardiovascular disease as the leading cause of mortality to deaths from cancer among adults aged 35–70 years [1]. Based on data from the International Agency for Research on Cancer (IARC), cancer incidence is supposed to increase from 19.3 million in 2020 to 30.3 million in 2040 worldwide for all age groups [2]. The IARC also reports that female breast cancer was the most commonly diagnosed cancer in 2020, surpassing lung cancer for the first time, followed by prostate cancer in third place [3]. An estimated worldwide increase of 1.5% annually in the number of new cases of female breast cancer, from 2.3 million in 2020 to 3.2 million in 2040, mainly due to demographic changes, is expected [2]. The incidence of prostate cancer is estimated to grow from 1.4 million in 2020 to 2.4 million in 2040 [2]. This increase will also affect regions with a high and very high human development index (HDI), where the estimation for breast cancer incidence rises from 1.83 to 2.3 million, and for prostate cancer, from 1.3 to 1.9 million from 2020 to 2040 [2]. Male breast cancer is a rather rare diagnosis, affecting less than 1% of all breast cancer cases, but with increasing incidence worldwide [4]. Moreover, its incidence is expected to increase further due to the spread of obesity, diabetes, and metabolic syndrome as prominent risk factors [5].

Against this background, a detailed exploration of therapeutic options is indicated, especially for these common sex-specific cancers.

### 1.1. Sex-Hormone-Dependent Malignancies

Most breast cancers and almost all initial prostate cancers are hormone-dependent in that their growth and proliferation depend on a specific nuclear receptor (NR) for androgen or estrogen [6].

In a French study with 2020 women, at least more than half of the cancers were estrogen-receptor-positive (ER+) with an increasing tendency and growing ER+ levels from perimenopause to postmenopause [7]. Regarding the progesterone receptor status, more cancers were receptor positive (PgR+) in premenopausal than in peri- or postmenopausal women, and levels of PgR+ decreased with menopausal status [7]. Additionally, the Nurses Health Study revealed that 77% of invasive female breast cancers were androgen-receptor (AR)-positive [8].

Although specific research is sparse, it is obvious that male breast cancer differs significantly from the female form, including a higher rate of ER+ tumor entities in the vast majority of reported cases. Additionally, androgen receptors are expressed in most male breast cancers [4]. In comparison, male breast cancer shows more parallels to female postmenopausal breast cancer than to premenopausal cancer, especially in terms of the age–frequency distribution [9]. However, the most common sex-hormone-dependent malignancy in men is prostate cancer, meaning that most are androgen-dependent, at least initially [10]. The hormonal dependency is of particular importance for treatment with androgen-deprivation therapy (ADT), primarily applied in locally advanced, recurrent, and metastatic stages of prostate cancer, but the AR frequently changes in the course of this therapy [11,12].

Since the interaction of sex hormones and their respective receptors is essential for the development of sex-hormone-dependent malignancies, female and male breast cancer and prostate cancer are frequently treated with anti-hormonal agents [13,14].

### 1.2. Side Effects, Quality of Life, and Adherence

Endocrine therapy has many side effects and therefore compromises patients’ quality of life in many dimensions and impacts their adherence to treatment [12,15,16].

According to a systematic review, adherence to adjuvant endocrine therapy among patients with breast cancer ranged from 41 to 72%, and discontinuation ranged from 31 to 73% after five years of treatment [17]. There have been reports of differences between men and women in adherence to tamoxifen, a commonly prescribed selective ER modulator in breast cancer, with men showing lower tolerance to side effects and a higher likeliness of discontinuation [5]. Reported side effects included sexual dysfunction, weight gain, hot flashes, and mood alterations [5]. Adjuvant endocrine therapy with tamoxifen or aromatase inhibitors (AIs) for early female breast cancer was found to be associated with several adverse effects involving vasomotor, musculoskeletal, and vulvovaginal function, as well as cardiovascular and cognitive dysfunction and fatigue [15]. Therefore, quality of life deteriorates, and adherence decreases substantially. The authors of this review conclude that common side effects are underestimated, and the resulting lack of adherence can lead to worsening treatment outcomes, which should be prevented by offering patients available therapeutic approaches that mitigate the toxicity [15].

ADT in prostate cancer patients has many side effects, which may compromise the quality of life and require additional medication [12]. For instance, ADT increases the risk of osteoporosis and cardiovascular disease, and metabolic disturbances can impair mental health and cognitive performance and lead to hot flashes, the most commonly complained side effect [12]. Novel non-steroidal anti-androgen agents for prostate cancer were found to be associated with increases in neurologic side effects such as falls and asthenia but not with a significant increase in treatment discontinuation due to adverse events [18]. Neoadjuvant hormone therapy worsened the side effects of radiotherapy or brachytherapy and compromised quality of life across multiple domains in a retrospective analysis of prostate cancer treatment outcomes [19].

### 1.3. Resistance to Anti-Hormonal Therapy and Rationale for a New Systematic Review

Sex-hormone-dependent cancers require specific NRs, which are the respective targets for anti-hormonal agents [6]. Therapy with the continuous administration of anti-hormonal agents in hormone-sensitive malignancies, above all prostate and breast carcinomas, can lead to the development of resistance in tumor cells and thus make the anti-hormonal therapy less effective [20,21]. In a study of postmenopausal women with advanced breast cancer, virtually all participants under the continuous application of tamoxifen developed resistance during the observation period [22]. In another study of advanced prostate cancer, the median time to relapse was 33 months in non-metastatic and 16 months in metastatic patients [23]. Therefore, the development of resistance is a major risk in endocrine therapy of primary-hormone-sensitive prostate and breast cancer. Several studies and reviews indicate that the intermittent use of anti-hormonal agents may be able to reverse or completely prevent resistance and is not inferior to the continuous application but may lead to fewer side effects, improved quality of life, and better compliance [24,25,26,27]. Especially for breast cancer, drug holidays in endocrine therapy could offer new therapeutic possibilities for reducing recurrence and mortality [28,29].

The introduction of new therapeutics and the paucity of systematic reviews, especially for breast cancer, make it necessary to conduct a new systematic review of intermittent endocrine therapy for sex-hormone-dependent malignancies to reflect the current state of knowledge and research. In addition to the primary outcomes of progression-free survival, disease-free survival, and overall survival as important cancer- and therapy-specific outcomes critical for decision making, the secondary outcome (time to) resistance to endocrine agents was included. Further secondary outcomes were cancer-specific survival, loco-regional control, side effects, adherence, and patient-reported quality of life, which are more patient-centered.

In preparing this protocol, a search of PROSPERO, the Cochrane Database of Systematic Reviews (CDSR), the Joanna Briggs Institute (JBI) Systematic Review Register, and the Open Science Framework (OSF) registries was conducted in December 2021 to determine if there were any previously registered protocols on the topic.

### 1.4. Aim and Objectives

The “Patient/Population, Intervention, Comparison/Control, Outcome, Study design” statement (PICOS) was applied to frame the research question.
**P** Patients with potentially sex-hormone-dependent malignancies receiving anti-hormonal therapy.**I** Intermittent or periodic administration of anti-hormonal substances (therapeutic intervals, drug holiday, alternating adherence).**C** Continuous administration of anti-hormonal substances.**O** Progression-free survival, disease-free survival, overall survival.**S** Randomized controlled trials, non-randomized controlled trials, pragmatic clinical trials, case–control studies, and comparative cohort studies.


Hence, we expect this systematic review to answer the following research question:

Are there any relevant differences in outcomes between intermittent and continuous endocrine therapy with anti-hormonal agents in potentially sex-hormone-dependent malignancies?

In addition to the primary research question, information on the quality of life, time to drug resistance, side effects, and adherence was considered. Finally, yet importantly, a comprehensive overview of the current scientific knowledge regarding intermittent ant-hormonal therapy for major sex-hormone-dependent malignancies is given.

## 2. Materials and Methods

The preparation of this systematic review protocol is based on the Preferred Reporting Items for Systematic Review and Meta-analysis Protocols (PRISMA-P) guidelines [30]. The respective checklist can be found as Appendix A (see Appendix A, PRISMA-P Checklist). As far as possible, the proposed systematic review follows the recommendations of the Cochrane Collaboration’s Handbook for Systematic Reviews of Interventions [31]. The protocol is registered at the International prospective register of systematic reviews PROSPERO (CRD42022312568, https://www.crd.york.ac.uk/PROSPERO/, accessed on 26 March 2022).

### 2.1. Eligibility Criteria

Inclusion and exclusion criteria applied to the titles, abstracts, and articles retrieved in the systematic literature search are depicted in detail in Table 1.

### 2.2. Search Strategy

An initial limited search based on the PICOS key elements and eligibility criteria, as reported above, was undertaken in PubMed, the Cochrane Library (CDSR and CENTRAL), and Embase (via OVID) to identify articles on the topic. The text words contained in the titles and abstracts of relevant articles and the database-specific index terms of the articles were used to develop a preliminary full search strategy for PubMed, with special consideration of the tree structure, and according to the PRESS specifications [32]. This search strategy, including all identified keywords and index terms as shown in Table 2, was customized for each database and/or information source to be searched. The electronic database search included PubMed, the Cochrane Library, Embase via Ovid, and Web of Science. The time period of publications to include was not limited. Sources of gray literature included OpenGrey via Data Archiving and Networked Services (DANS), Pro-Quest for dissertations and theses, and ClinicalTrials.gov and the Cochrane Library (CENTRAL) for unpublished studies. The reference list of all included publications was screened for additional studies. In order to obtain the greatest possible completeness, the systematic reviews found in the literature search were checked to see whether the included studies were included in our reference list or could still be included subsequently.

### 2.3. Data Management and Selection

After completion of the search process, all identified references were compiled and uploaded into RefWorks, automatically checked for duplicates, and the duplicates were removed. After a pilot test of 25 randomly selected titles/abstracts and a review and refinement or adjustment of the predefined inclusion and exclusion criteria, the titles and abstracts were screened by two independent reviewers using the final inclusion and exclusion criteria (Table 1). The publications that scored positively in this title/abstract scan were included and subsequently obtained in full text, which was also evaluated in detail by two independent reviewers according to the inclusion/exclusion criteria. The reasons for excluding references that did not meet the inclusion criteria at the title/abstract and full-text level were documented and provided in the systematic review report. Any disagreements that may arise between the reviewers at any stage of the selection process will be resolved through discussion or by a third reviewer. The results of the search and the inclusion and exclusion procedure of all references will be fully reported in the final version of the systematic review and presented in a PRISMA flow diagram [33].

### 2.4. Data Extraction and Risk of Bias Assessment

Data from the references included in the selection process were extracted by one researcher (J.K.) into a predefined data extraction table, which was developed and tested in advance and controlled by a second (D.K.). The extracted data contained specific information related to the PICOS elements, study methods, and key outcomes relevant to the objectives of the review. The customized data extraction table is divided into categories of content-linked fields related to specific data regarding publication, study design and participants, intervention and control, and overall assessment. This draft extraction form, which can be modified and updated as needed during the process of data extraction from each included source, is provided (see Appendix A). Any changes to this version were delineated in the final report of the systematic review. Disagreements that arise between the reviewers during the process of data extraction will be resolved by discussion or another reviewer, as before. If necessary, further information will be sought in other publications of the particular study or in a published protocol, or the authors of the articles will be contacted to obtain missing or additional data.

To determine the quality of the evidence of included references, a comprehensive risk of bias assessment was conducted. Following a joint training session based on the knowledge of an experienced reviewer, the Cochrane Risk of Bias 2 (RoB 2) Tool was applied for RCTs [34], and the ROBIN-I tool was applied for non-randomized studies [35]. The RoB 2 assesses bias in five domains leading to an overall risk of bias judgments of “low”, “some concerns”, or “high risk” for RCTs [34]. The ROBINS-I tool for non-randomized studies includes seven domains and a judgment of the respective risk of bias, which is carried forward to an overall risk of bias from “low” to “critical” and “no information” [35].

### 2.5. Data Analysis and Presentation

A comprehensive description and the PRISMA flow diagram provide a detailed overview of the selection process of all included sources. To reach the overarching aim of giving a comprehensive overview of the current scientific knowledge regarding intermittent anti-hormonal therapy, the data are presented in a hierarchically structured manner. On the first level, descriptive statistics and their graphical representations will be applied to quantify the distribution of publications across cancer entities and, if required, sub-sections. The second level includes the different study designs, e.g., RCTs and non-randomized trials. Results will be presented in summary tables that include more detailed information on specifics of each study, outcomes, conclusions, and risk of bias. Further information will be given in a narrative form. For RCTs, the feasibility of a meta-analysis will be assessed and, if enough studies with sufficient data are available, this will be completed while taking into account the results of the risk of bias assessment for eligible studies, i.e., those RCTs published in the period between the latest inclusion date of the most recently published meta-analysis and the final date of our systematic literature search. In particular, with regard to ADT in prostate cancer, such a meta-analysis seemed to be possible due to the more frequent use of intermittent therapy regimens and the existence of previous meta-analyses [26,36]. Sufficient data provided, a meta-analysis will be performed for the primary outcomes of overall survival and progression-free survival. Hazard ratios (HRs) and their respective 95% confidence intervals (CIs) regarding intermittent ADT versus continuous ADT will either be extracted from the included studies or estimated from published data [37]. Based on homogeneity of study results, either a fixed-effect model with inverse variance-weighted confidence intervals or a random-effects analysis with weighted method is applied. The I^2^ statistic will quantify heterogeneity, and a sensitivity analysis has been planned considering the risk of bias estimates and studies with results on the margins of the result field. A funnel plot will provide an estimate of the publication bias for included studies. Finally, the quality of the body of evidence will be assessed by applying the specific evidence grading system of the GRADE Working Group [38]. The meta-analysis will be performed using the package “metafor” for R (https://cran.r-project.org/, accessed on 30 March 2022) according to the manual of Viechtbauer [39].

## 3. Handling the Results

The presentation of results follows the PRISMA statement for systematic reviews [33]. Finally, any deviation from this protocol is reported with an explanation in the publication of the results.

### Dissemination of Results

The dissemination of the results takes place in a variety of ways, foremost through publication in an open access journal, but also through conference proceedings and lectures. In addition, this systematic review shall serve to open up new research foci with regard to intermittent anti-hormonal therapy: where there is a lack of data (e.g., HRQoL, adherence), where can the use of intermittent therapy usefully replace or extend the continuous use, how the use of intermittent therapy can be offered to reduce the number of treatment dropouts, and how the cost-effectiveness compared to continuous use.

## 4. Discussion

Although ethical approval is not required for this work, ethical medicine obligates us to put the patient’s welfare first and carefully consider therapeutic alternatives and compare them to current standards. This is especially the case when standard procedures have many side effects and therefore impact the quality of life.

In addition to investing in new therapeutics, science should also focus on the application of existing and proven therapies. Overall survival and disease-specific survival are the most important measures underlying cancer research. Proven therapies are predominantly well-studied in this regard; therefore, therapeutic targets focusing on patient well-being can now be addressed. These primarily include patient-related outcomes such as HRQoL, which can also be assumed to interact with the side-effect profile to affect treatment adherence.

The intermittent application of anti-hormonal agents has several possible advantages, such as fewer side effects, improved health-related quality of life, and better adherence [24,25,26,27]. While little is known to date about intermittent therapy in breast cancer, most current knowledge comes from studies on prostate cancer. For better adherence, intermittent application of endocrine therapy should be considered, especially in breast cancer patients. Rates of discontinuation regarding anti-hormonal therapy in women with breast cancer reach levels as high as 73%, necessitating an increased search for alternative therapy regimens [17,25]. Although the presumably only currently known randomized controlled trial with intermittent administration of aromatase inhibitors failed to demonstrate significant differences in adverse effects, the absence of evidence should not be taken as evidence of absence.

The systematic review presented in this protocol is designed to identify the current scientific evidence on intermittent anti-hormonal therapy in the most prevalent sex-specific malignancies, which pose an immense burden on public health and healthcare systems worldwide [40]. It is impossible to present a complete picture of the available information on this subject, so it must be assumed that this review can only present a limited excerpt. The focus on RCTs in high-quality systematic reviews cannot reflect the situation in everyday life that affects millions of people with cancer. However, a review can provide evidence-based information that can then be integrated into everyday decisions and improve care for affected people in a more patient-centered way.

## 5. Conclusions

Both breast cancer and prostate cancer are predominantly dependent on sex hormones for their development. Anti-hormonal therapy often leads to the emergence of resistant cancer cells in both prostate and breast cancer. While intermittent endocrine therapy has found its way into the guidelines for the treatment of prostate cancer, there have been no efforts to date in this regard for the treatment of breast cancer. This systematic literature review intends to provide as comprehensive a picture as possible of the respective current state of research and, at best, to encourage an impetus for further research, particularly in the area of breast cancer.

## Figures and Tables

**Table 1 ijerph-19-15486-t001:** Structured overview of inclusion and exclusion criteria according to the “Patient/Population, Intervention, Comparison/Control, Outcome, Study design” (PICOS) approach.

	Inclusion	Exclusion
Patient/Population	Human participants (adults ≥ 18 years of all sexes) with a confirmed diagnosis of breast or prostate cancer at any stage, with or without known resistance to anti-hormonal therapy and without restriction to previous or current application of other therapeutic agents.	Children, animal, or in vitro testing.
Intervention	Intermittent or periodic application of anti-hormonal therapy targeting sex-specific hormones, their respective receptors, releasing hormones or enzymes for the synthesis from precursors: agonists, antagonists, inhibitors, receptor modulators, and degraders, anti-hormonal agents. An intermittent or periodic application should include at least one break (tb) of either predetermined length or with fixed criteria to stop and/or restart the application (ta). For breaks of predetermined length, ta ≥ tb is applied. Reports of intermittent application due to failure of adherence are not excluded but handled separately. An observation period of at least 18 months is required.	Non-sex-hormone-targeted therapeutics: chemotherapeutic agents, immunotherapeutics, mTOR inhibitors, HER2-therapeutics, CDK4/6 inhibitors, PARP inhibitors, etc.
Comparison/Control	Continuous application of anti-hormonal therapy.	
Outcome	Primary outcomes: progression-free survival, disease-free survival, overall survival. Secondary outcomes: cancer-specific survival, loco-regional control, health-related quality of life (HRQoL), time to drug resistance, side effects, adherence.	Studies will not be excluded due to missing outcomes, but the results will be reported separately.
Study types/ Publication	Randomized controlled trials (RCTs), cluster RCTs, non-randomized controlled trials, case–control studies, controlled before–after studies, longitudinal comparative cohort trials (observational study design). Publication language is restricted to English and German. No restriction to the time period of publication.	Cross-sectional studies, case reports, case series, qualitative studies, modeling studies.
Setting	No restrictions to types of settings.	

**Table 2 ijerph-19-15486-t002:** PubMed search strategy based on the PICOS statement and inclusion/exclusion criteria, executed on 12 January 2022.

Step	Search String	Results
#1	“neoplasms, hormone dependent/drug therapy”[MeSH Terms:noexp] OR “prostatic neoplasms/drug therapy”[MeSH Terms] OR “breast neoplasms/drug therapy”[MeSH Terms]	96,675
#2	“drug resistance, neoplasm”[MeSH Terms] OR “androgen antagonists/therapeutic use”[MeSH Terms] OR “gonadotropin releasing hormone/agonists”[MeSH Terms] OR “estrogen antagonists/therapeutic use”[MeSH Terms] OR “estrogen receptor modulators/therapeutic use”[MeSH Terms] OR “anticarcinogenic agents/therapeutic use”[MeSH Terms] OR “antineoplastic agents, hormonal/therapeutic use “[MeSH Terms] OR “antineoplastic agents, hormonal/antagonists and inhibitors”[MeSH Terms] OR “chemoprevention”[MeSH Terms:noexp]	97,743
#3	#1 and #2	22,161
#4	“clinical protocols”[MeSH Terms] OR “drug administration schedule”[MeSH Terms] OR “drug tapering”[MeSH Terms] OR “time factors”[MeSH Terms] OR “patient compliance”[MeSH Terms]	1,545,304
#5	“intermittent*”[Title/Abstract] OR “sequent*” [Title/Abstract] OR “periodic*”[Title/Abstract] OR “drug holiday*”[Title/Abstract] OR “therapy persistence”[Title/Abstract]	418,847
#6	#4 or #5	1,926,728
#7	“treatment failure”[MeSH Terms:noexp] OR “survival analysis”[MeSH Terms] OR “recurrence”[MeSH Terms:noexp]	534,832
#8	#3 and #6 and #7	1445
#9	#8 AND ((classicalarticle[Filter] OR clinicaltrial[Filter] OR controlledclinicaltrial[Filter] OR guideline[Filter] OR meta-analysis[Filter] OR observationalstudy[Filter] OR pragmaticclinicaltrial[Filter] OR randomizedcontrolledtrial[Filter] OR review[Filter] OR systematicreview[Filter]) AND (humans[Filter]) AND (english[Filter] OR german[Filter]) AND (alladult[Filter]))	583

Note. MeSH medical subject heading, # number.

## Data Availability

Not applicable.

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
