# Peer review of "Intermittent Use of Anti-Hormonal Agents for the Endocrine Therapy of Sex-Hormone-Dependent Breast and Prostate Cancer: A Protocol for a Systematic Review"

_ijerph, 2022, doi:10.3390/ijerph192315486_

Round 1

Reviewer 1 Report (Previous Reviewer 1)

no comments.

Author Response

Thank you for reviewing our manuscript!

Reviewer 2 Report (New Reviewer)

The authors have collected most of the relevant literature with respect to both the cancer and presented very well as a systemic review.

1. In the Discussion and Conclusion section , authors could have presented the following observations.

a) The advantages of Intermittent Use of Anti‐Hormonal Agents

b) Does sex play a vital role? Who responds well to the therapy? Male or Female?

c) Does age play any role?

2. The Important findings or observations could have been given as a bulleted point for better understanding of common readers.

Author Response

Thank you for your assessment and suggestions to improve our manuscript.

We ask for your understanding that we can only address these points in detail in our report after completion of the systematic review and not in the protocol presented here. Nevertheless, we have added a paragraph to our discussion regarding your proposal.

Discussion, page 24, line 277-287:

The intermittent application of anti-hormonal agents has several possible advantages like less side-effect, improved health-related quality of life and better adherence [24-27]. While little is known to date about intermittent therapy in breast cancer, most current knowledge comes from studies in prostate cancer. For better adherence, intermittent application of endocrine therapy should be considered especially in breast cancer patients. Rates of discontinuation regarding anti-hormonal therapy in women with breast cancer reach levels as high as 73%, necessitating an increased search for alternative therapy regimens [25,40]. Although the presumably only currently known randomized controlled trial with intermittent administration of aromatase inhibitors failed to demonstrate significant differences in adverse effects, the absence of evidence should not be taken as evidence of absence.

Reviewer 3 Report (New Reviewer)

Schematic or flowchart (model) will strengthen the manuscript

Author Response

Thank you for reviewing our protocol.

Please understand that we are unable to produce a flowchart at the current stage of the systematic literature review. We will include this in the final report of our systematic review.

This manuscript is a resubmission of an earlier submission. The following is a list of the peer review reports and author responses from that submission.

Round 1

Reviewer 1 Report

The manuscript entitled “Intermittent Use of AntiHormonal Agents for the Endocrine Therapy of Sex HormoneDependent Breast and Prostate Cancer: A Protocol for a Systematic Review” aimed to summarize and evaluate the current knowledge regarding intermittent endocrine cancer therapy, such as breast and prostate cancer. Although the experimental method of the manuscript conformed to the standard, the innovation of the research was general, and the presentation of the results was too simple. There was no detailed analysis of the research content in the discussion part, and it had limited significance for the existing treatment guidance.

Reviewer 2 Report

The authors provided sufficient background information regarding hormone-specific cancers, including the cause and concerns on endocrine therapies. Then they proposed a detailed method for systematic review of intermittent endocrine intervention strategy on such cancers. However, first for the protocol, the authors did not mention the specific drugs used. This can be very misleading if you do not mention or simply mix the dose and the specific component used. Second, the physiological conditions of patients are also very important. Whether they have any complications or other physiological issues, whether they smoke/drink, and at which stage they have the cancers are of vital importance for the consideration of therapeutic effects. Except for these, the biggest flaw of this protocol is its application. Different from a protocol that can be applied commonly to an experiment, this protocol has very limited use, which made this manuscript provided little useful information. Why not use this method to just write out a systemic review on this topic? 

The main question addressed by the research is to develop a protocol for a systematic review to compare the outcomes between intermittent and continuous endocrine therapy with anti-hormonal agents in potentially sex-hormone dependent malignancies. Indeed the authors developed a detailed protocol, which is original in the field, but it added few value to the field, as the application of such protocol is very limited, and the protocol per se has critical flaws (as mentioned in the submitted comments).

Reviewer 3 Report

I went through the manuscript "Intermittent Use of Anti‐Hormonal Agents for the Endocrine 2 Therapy of Sex Hormone‐Dependent Breast and Prostate    3 Cancer: A Protocol for a Systematic Review"

This manuscript is well prepared the search string is correct and  the inclusion and exclusion criteria are well selected.

The message of this review will be important for oncologists to make a correct  decision

Reviewer 4 Report

The authors of this protocol intend summarize the available high-quality evidence (i.e. RCT) on the intermittent application of antihormonal treatment in two large cancer entities, breast and prostate cancer. This work is well structured and its results are very valuable since they may have both clinical and public health care consequences. Hence I believe that the protocol should be published after a few minor changes. The final paper will – hopefully – be available soon.

·         Endpoints: I agree that OS and PFS are the most important endpoints. Nevertheless, I would suggest to include cancer specific survival (CSS) and loco-regional control (LRC) as well, since they are good endpoints for therapy response.

·         Conclusions: At the present stage, the authors do not present any results on which conclusion could be drawn. Therefore it seems to me that Conclusions is the wrong title for the last section, it should rather be something like Potential findings.

A quick Pubmed search revealed 250 systematic reviews for the terms  breast cancer and endocrine therapy and 27 for the terms prostate cancer and endocrine therapyt. It does not reveal a single systematic review for the term intermittent endocrine therapy.

As stated above, the evidence on the intermittent use of endocrine treatment has not been summarized systematically. Hence, the proposed systematic review fills this knowledge gap.

In fact, I cannot detect any methodological flaws. The protocol was set up according to the PRISMA and PROSPERO guidelines, which are considered state of the art for systemtatic reviews. A search in the following databases was performed to find out whether there was already a systematic review on the topic: PROSPERO, Cochrane Database, JBI and OSF – obviously the result was negative, which means that the proposed work really fills a knowledge gap.

At this stage of the analysis (i.e. protocol) the references are appropriate.

Tables 1 and 2 describe in detail the inclusion / exclusion criteria and the search strategy both of which are well understandable and clear to the reader. This protocol does not contain any figures